# Nano TiO_2_ Imparting Multifunctional Performance on Dyed Polyester Fabrics with some Disperse Dyes Using High Temperature Dyeing as an Environmentally Benign Method

**DOI:** 10.3390/ijerph17041377

**Published:** 2020-02-20

**Authors:** Alya M. Al-Etaibi, Morsy Ahmed El-Apasery

**Affiliations:** 1Natural Science Department, College of Health Science, Public Authority for Applied Education and Training, Fayha 72853, Kuwait; 2Dyeing, Printing and Textile Auxiliaries Department, Textile Research Division, National Research Centre, 33 El Buhouth St., Dokki, Cairo 12622, Egypt; elapaserym@yahoo.com

**Keywords:** high temperature dyeing, self-cleaning, TiO_2_ NPs, ultraviolet protection factor

## Abstract

Polyester fabrics were dyed with prepared disperse dyes using the high temperature dyeing method. The dye exhaustion of the dye baths were compared to the low-temperature dyeing method in an attempt to study the proportion of the dye effluent solution that affects the environment. The dye uptake of the high temperature dyeing method (HT) of polyester fabric was compared with low temperature dyeing, hence (HT) increased the color strength of the investigated dyes by 309 and 265%. This means that the amount of dye present in the dye effluents by using the high-temperature dyeing method is almost non-existent, and this is reflected positively on the environment as these wastes pollute the environment. Post-treated polyester fabric was prepared through a two-step hot process after being immersed in a solution of Titanium (IV) oxide nanoparticle sizeTiO_2_ NPs (21 nm primary particle size) at 80 °C followed by curing at 140 °C. The treated fabric realized an optimum UV protection factor of 34.9 and 283.6 degrees. These fabrics also demonstrated a strong ability to improve the light fastness properties. Finally, the potential applications of such value-added fabrics as self-cleaning and antifungal activities were investigated. The results indicated that the treated dyed fabrics with TiO_2_ NPs endowed fabrics with the excellent self-cleaning of methylene blue dye. From the above, the treated fabrics with nano-titanium dioxide can be used in some promising fields, for example, medical ones.

## 1. Introduction

High temperature dyeing is the most extensive strategy for batch coloration, grants increased diffusion, and along these lines, the expanded rate of dyeing by lessening the union between the polymer chains increasing the kinetic energy of the dye molecules [1,2,3]. Environmental problems are related to the dye baths utilized with a wide scope of various components added to the dye bath, regularly at high concentrations. The textile industries faced requirement of treatment a large part of the industrial waste water from dyebaths. Dyeing at 130 °C makes swelling of the fiber even and dye molecules penetrate the fiber polymer more than that accomplished in the low temperature dyeing method at 100 °C. Utilizing nanotechnology has upgraded the functions of material to multifunctional fabrics with antibacterial and UV protection properties and stain resistance have been reported [4,5,6,7,8,9]. Titanium (IV) oxide nanoparticle sizeTiO_2_ NPs are better than different photo-catalysts because of their reasonable safety, high photo-catalytic property, and effectiveness. Recently, many studies have reported on the possibility of titanium dioxide NPs as a photo-catalyst to give polyfunctional merits to polyester fabrics [10,11,12,13]. Improvement of polyester fabric by using TiO_2_ NPs to provide it with multifunctional properties (e.g., UV-resistance, stain resistance, and antibacterial) was one of the most important goals of this study. Generally, the study here was divided into two main parts: the first part was to study the dyeing performances of some disperse dyes using high temperature as an environmentally benign method where colored dye effluents of the remaining solution of dye baths affects the environment and damages it; the second part is promoting the fabric with multiple functions after dyeing (post dyeing with disperse dyes) by using TiO_2_ NPs (21 nm primary particle size) through a two-step hot process after being immersed in a solution of TiO_2_ NPs at 80 °C followed by curing at 140 °C.

## 2. Materials and Methods 

### 2.1. Materials 

#### 2.1.1. Fabric

Scoured and bleached 100% polyester fabric (149 g\m^2^) was supplied by the El-Mahalla El-Kobra Company. The fabric was scoured in an aqueous solution with a liquor ratio1:30 containing 2 g/L of nonionic detergent solution (Hostapal, Clariant) and 2 g/L sodium carbonate at 50 °C for 30 min to remove impurities.

#### 2.1.2. Titanium (IV) Oxide TiO_2_ NPs

Nanopowder, 21 nm primary particle size (Tem), ≥ 99.5% trace metals basis, Lot # MKBV3126V was purchased as 718467-100G from ALDRICH Chemistry.

### 2.2. Methods

#### 2.2.1. Dyeing at 130 °C (High Temperature)

In this dyeing process, which was conducted at a temperature of 130 °C using the Infra Red machine as a source of heat, the dye weight was calculated at 2% shade (w.o.f). This quantity was placed with weighted polyester fabric in a cup, then the calculated amounts of the dye and Matexil DA-N as the dispersing agent were dissolved in some drops of dimethylformamide solvent, and water was added to reach 1:50 of liquor ratio. The pH was adopted at 4.5 and the dyeing process was conducted for an hour. The dyed samples underwent a reduction clearing process for 20 minutes at 60 °C.

#### 2.2.2. Treatment of Polyester Dyed Fabrics by TiO_2_ NPs

Treatment of polyester dyed fabric by nano-titanium dioxide was conducted through the exhaustion method, where the fabric was treated with different concentrations of nano-titanium dioxide ranging from (1%) to (5%) for 20 minutes at 80 °C and the liquor ratio was 1:30. The wet fabric was first squeezed to remove the excess dispersion and then the fabric was dried for 15 minutes at 70 °C. The dry fabric was cured for a second timefor 10 minutes at 140 °C and then the fabric was washed at 60 °C for 20 minutes in an aqueous solution with a liquor ratio of 1:30 containing 3 g/L nonionic detergent solution (Hostapal, Clariant). Finally the treated fabric was left to dry.

### 2.3. Color Strength

The colorimetric analysis of the dyed samples was performed using a Hunter Lab Ultra Scan PRO spectrophotometer.

K/S values expressed the color strength, which was performed by utilizing the equation of Kubelka–Munk [14].
K/S = [(1 – R)^2^ / 2R] – [(1 – R_o_)^2^/2R_o_]
where R is the decimal fraction of the reflectance of the dyed fabric; R_o_ is the decimal fraction of the reflectance of the not dyed fabric; K is the absorption coefficient; and S is the scattering coefficient.

### 2.4. Fastness Property Tests

The fastness properties of the dyed samples like perspiration, washing, light, and rubbing were tested according to the tests of the American Association of Textile Chemists and Colorists [15].

#### 2.4.1. Washing Fastness 

The composite examples were sewn between two bits of dyed cotton and wool fabrics and afterward drenched in an aqueous solution containing 5 g/L of nonionic detergents at 60 °C for 30 min. Samples were removed and dried. Assessment of the wash fastness was set up utilizing the grey scale for color change [15].

#### 2.4.2. Rubbing Fastness

##### Dry Rubbing

The test specimen was placed flat on the base of the crock meter and a white testing fabric was mounted. A covered finger was lowered onto the test specimen and slid back and forth 20 times. The white test sample was then removed for evaluation using the grey scale for staining [15].

##### Wet Rubbing

The white test sample was thoroughly (65%) wet with water. The procedure was run as before. The white test samples were air dried before evaluation [15].

#### 2.4.3. Perspiration Fastness

The acidic solution was prepared by dissolving L-histidinemonohydrochloride monohydrate (0.5 g), sodium chloride (5 g), and sodium dihydrogen orthophosphate dihydrate (2.2 g) in one liter of water, and the pH was adjusted to 5.5. On the other hand, the alkaline solution was prepared by dissolving L-histidinemonohydrochloride monohydrate (0.5 g), sodium chloride (5 g), and disodium hydrogen orthophosphate dihydrate (2.5 g) in one liter of water, and the pH was adjusted to 8. The colored specimen was sewn between two pieces of uncolored specimens. The composite samples were immersed for 30 min in both solutions. The test specimens were placed between two plates of glass under a force of 5 kg in an oven at 37 ± 2 °C for 4 h. The effect on the color of the tested specimens was expressed and defined by reference to the grey scale [15].

#### 2.4.4. Light Fastness

This test was carried out by utilizing a xenon lamp and continuous light for 35 h. The effect on the color of the tested samples was recorded through reference to the blue scale for color change [15].

### 2.5. Photo-Stimulated Color Removal on Polyester

A total of 0.01 g/L of methylene blue was marked on both the post-treated TiO_2_ NPs (1–5%) treated polyester and the untreated fabrics. The polyester fabrics were illuminated through exposure to an ultraviolet lamp for 12 hours.

### 2.6. Ultraviolet Protection Factor Measurement 

It is worth noting that the ultraviolet protection factor is the capability of dyed polyester fabric to block ultraviolet, which was conducted in an ultraviolet visible spectrophotometer 3101.

### 2.7. Antimicrobial Activity Test

The antimicrobial activities of the dyed fabrics with disperse dyes 1 and 2 were tested using the agar-well diffusion technique against six different microbial cultures. *Bacillus cereus* and *Staphylococcus sciuri* (Gram-positive bacterium); *Escherichia coli* and *Pseudomonas aeruginosa* (Gram-negative bacterium); *Aspergillus flavus* and *Penicillium chrysogenum* (fungi) were used in the test. The published method was followed when evaluating antimicrobial activities [15].

### 2.8. Evaluation of Antimicrobial Activities

The antimicrobial activities were determined at the Friendly Human Bacterial Unit, National Research Centre, Cairo, Egypt.

## 3. Results and Discussion 

Some disperse dyes have recently been reported by us (Figure 1) [14,15]. These disperse dyes were utilized for polyester dyeing with 2% shade (weight of fabric). The applied dyeing method was high temperature (HT) at 130 °C, deep greenish yellow, and orange yellow colors were obtained.

### 3.1. Dye Uptake 

The polyester dyed fabrics were surveyed using a tristimulus colorimeter, where L* represents lightness and (C) represents the chroma. The listed data in Table 1 reveals that the hue of the dye expressed as (*h*) values show that practically the entirety of colored polyester fabrics communicated a similar hue. The positive estimations of *b** demonstrated that the color hues of the dyed polyester fabrics moved to a reddish direction. Color strength (K/S) represented the dye uptake and the colorimetric parameter values obtained for the high temperature and low temperature colored polyesters are listed in Table 1, which reveals that the high temperature dyed fabrics were darker than the low temperature dyed fabrics. The K/S values were19.38, 12.63, and 4.74 and 3.46 for the high and low temperature dyed fabrics, respectively. These outcomes demonstrate that the dye uptake of the high temperature dyed fabrics was more noteworthy than the low temperature dyeing method by 309% and 265%, respectively, so the high temperature dyeing method could be considered as an environmentally benign method that is able to reduce the pollution load in the colored dye effluents, which would otherwise have negatively affected the environment.

Moreover, the dyed fabrics at high temperature were accompanied by a rise in pressure as the rate of dye penetration inside the filament increased. It may be that the temperature helps to increase the kinetic energy of the dye molecules, and also that the temperature swells the polyester fibers, so the rate of dyeing is higher than the low temperature dyeing process.

### 3.2. Fastness Properties

Disperse dyes 1 and 2 were used for dyeing polyester fabrics via the high temperature dyeing method, Table 2 shows the excellent fastness data to rubbing, washing, perspiration and light fastness, which was slightly less than that displayed as very good for the untreated polyester dyed fabrics.

#### 3.2.1. Light Fastness of Both Untreated and Post-Treated Polyesters with TiO_2_ NPs

The light fastness of both the untreated and post-treated polyesters with TiO_2_ NPs of the dyed with disperse dyes were estimated after exposure of 35 hours by utilizing blue scale, which gives increasingly critical outcomes. Table 3 reveals that the use of the TiO_2_ NPs turned out to be more viable in the treated polyester fabrics than in the untreated samples. Additionally, the light fastness of the treated polyester with dye 1 was higher than the treated samples of disperse dye 2.

### 3.3. Self-Cleaning of Post-Treated Polyester

The most important capabilities of the nanoparticle post-treated fabrics is changing retained light to self-cleaning substances to remove its dirty [16]. The listed data in Table 3 reveals the impact of methylene blue stain on both untreated and nanoparticles post-treated polyester fabrics following 12 hours UV-illumination. A removal of methylene blue stain prompted by UV-light was observed for TiO_2_ NPs - treated polyester fabric reached to 80%. Treatment of polyester with TiO_2_ NPs is directed to the construction slight layer of TiO_2_ NPs on polyester surface. The high debasement impact of TiO_2_ NPs is showed up on the treated polyester. The surfaces that come about because of treating of polyester with TiO_2_ NPs may clarify its self-cleaning power. The hydrophobic surfaces can eliminate the dirty as methylene blue from the polyester dependent on various systems. Producing hydrophobic surfaces was created dependent on Lotus impact. The hydrophobic surface prevents the adsorption of dirt [17], keeping the surface of polyester clean all the time.

### 3.4. Ultraviolet Protection Factor (UPF) of Dyed Polyester Fabrics

Estimation of the ultraviolet protection factor [18,19] is dependent on the ultraviolet properties of disperse dyes 1 and 2. A consequence of Table 4 shows the increasing color strength of the dyed polyester untreated or post-treated with TiO_2_ NPs joined with the growing ultraviolet protection factor of the polyester fabrics. After dyeing, it will generally be assumed that polyester fabrics that had very good to excellent UV protection without treatment were 25.5 or 236.2. The listed data in Table 4 shows that the blank polyester fabric lacked an UPF value (8.2), the T (UV-A) value was 35.7, which means that this fabric does not provide any protection against ultraviolet light. Then again, the dyed polyester fabric with disperse dye 2 treated with TiO_2_ NPs (1%–5%) had an UPF value of 34.3–34.9 and the T (UV-A) values were higher than 6% (14.3–17.3), which imply that these fabrics had very good protection against UV radiation. Table 4, shows in reality, that the dyed polyester fabrics with disperse dye 1 treated with TiO_2_ NPs (1%–5%) had an UPF value of 255.3–283.6 and the estimation of T (UV-A) are under 1% (0.6–0.8), which imply that these fabrics have excellent protection against the radiation of ultraviolet.

### 3.5. Antimicrobial Activities

Untreated colored fabrics with dyes 1 and 2 using the high temperature dyeing method were tested for their inhibitor influence on the growth of two fungi *Aspergillus flavus* and *Penicillium chrysogenum* and two pathogenic G+ bacterial *B. cereus* and *S. sciuri*, pathogenic G− bacterial *P. aeruginosa* and *E. coli*. Natamycin served as the standard of fungi and Ceftriaxone was the standard of bacteria. The antimicrobial screening data listed in Table 5 showed that the investigated colored fabrics of dyes that didnot possess antimicrobial properties against two pathogenic fungi *Aspergillus flavus* and *Penicillium chrysogenum*. Additionally, the untreated colored fabric of dye 1 did not possess antibacterial properties against the four pathogenic bacterial strains, while the untreated polyester dyed fabrics of disperse dye 2 possessed very good antibacterial properties against *Staphylococcus sciuri*. Untreated polyester dyed fabrics of disperse dye 2 possessed strong antibacterial properties against *Pseudomonas aeruginosa* and very strong antibacterial properties against *Escherichia coli*, which wasa great result. Then again, the photo-catalytic impact of TiO_2_ NPs and inorganic nano-metal oxides is the primary purpose behind their antimicrobial action through the creation of active oxygen species, for example, hydrogen peroxide, super oxide anions, hydroxyl radicals, and singlet oxygen, which leads to the death of the bacterial cell [20]. The antifungal screening data listed in Table 6 showed that the treated polyester dyed fabrics with TiO_2_ NPs of disperse dye 2 did not possess antifungal properties against the two pathogenic fungi *Aspergillus flavus* and *Penicillium chrysogenum*. In contrast, the treated polyester dyed fabrics with TiO_2_ NPs of disperse dye 1 possessed strong antifungal properties against *Aspergillus flavus* and *Penicillium chrysogenum*. This is consistent with studies that have previously been published [21] where the treatment of polyester fabrics with inorganic NPs oxides may give these fabrics an antimicrobial function.

## 4. Conclusions

Disperse dyes were used for polyester dyeing through the high temperature method. These polyester fabrics gave great results for rubbing, washing, perspiration fastness, and very good fastness to light. The TiO_2_ NPs post-dyeing treatment is an easy method and is promising for producing substrates that possess good self-cleaning, light fastness, anti-UV, and anti-fungal properties.

## Figures and Tables

**Figure 1 ijerph-17-01377-f001:**
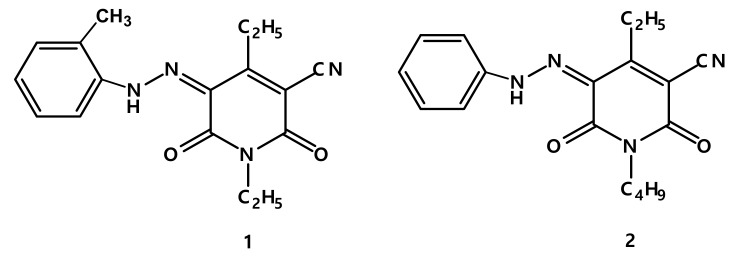
Disperse dyes 1 and 2.

**Table 1 ijerph-17-01377-t001:** Color strengths of the untreated polyester fabrics.

DyeNo.	K/S	λ_max_	L*	*a**	*b**	C	*h*
**High Temperature Dyeing at 130 °C**							
**1**	19.38	445	76.33	4.47	93.78	93.88	87.27
**2**	12.63	450	78.41	−0.56	87.34	87.34	90.37
**Low Temperature Dyeing at 100 °C** [15]							
**1**	4.74	445	80.06	−4.58	68.35	68.51	93.83
**2**	3.46	450	80.35	−5.20	61.82	62.04	94.80

**Table 2 ijerph-17-01377-t002:** Fastness properties of the dyed polyester by utilizing the high temperature dyeing method.

Dye No.	Rubbing Fastness		Washing Fastness			Light Fastness	Perspiration Fastness					
							Alkaline		Acidic			
	Wet	Dry	SC	SW	Alt		SC	SW	Alt	SC	SW	Alt
**1**	5	5	5	5	5	5-6	5	5	5	5	5	5
2	5	5	5	5	5	5	5	5	5	5	5	5

SW = Staining on wool, Alt = Alteration, SC = Staining on cotton.

**Table 3 ijerph-17-01377-t003:** Light fastness and Self-cleaning of the untreated and treated polyester fabrics with TiO_2_ NPs.

Dye No.	TiO_2_ %	Light Fastness	Self-Cleaning of Methylene Blue Stain
	untreated	5–6	No Removal
	1	5–6	80%
1	2	6	80%
	3	6	75%
	4	6	80%
	5	6	60%
	untreated	5	10%
	1	5	80%
2	2	5	75%
	3	5	80%
	4	5	70%
	5	5–6	65%

**Table 4 ijerph-17-01377-t004:** The ultraviolet protection factor (UPF) values and UPF categories with relative transmittance and protection levels of the untreated and treated polyester fabrics with TiO_2_ NPs.

Dye No.	TiO_2_ %	UPF	UPF Extent	UPF Valuation Protection	Transmittance	
					UV-A 315–400 nm	UV-B 290–315 nm
	Blank	8.2	<15	Insufficient	35.7	8.0
	untreated	236.2	40–50, 50+	Excellent	0.60	0.4
	1	255.3	40–50, 50+	Excellent	0.70	0.3
1	2	201.6	40–50, 50+	Excellent	0.80	0.4
	3	283.6	40–50, 50+	Excellent	0.50	0.3
	4	278.7	40–50, 50+	Excellent	0.60	0.3
	5	244.6	40–50, 50+	Excellent	0.60	0.3
	untreated	25.5	25–39	Very good	17.3	2.0
	1	34.3	25–39	Very good	14.4	1.3
	2	34.0	25–39	Very good	15.0	1.2
2	3	34.9	25–39	Very good	14.3	1.2
	4	32.6	25–39	Very good	14.8	1.3
	5	32.5	25–39	Very good	15.0	1.3

**Table 5 ijerph-17-01377-t005:** Antimicrobial activity of the untreated polyester dyed fabrics.

Dye No.	Inhibition Zone Diameter (Nearest mm)					
	G^+^ Bacteria		G^−^ Bacteria		Fungi	
	*Bacillus* *cereus*	*Staphylococcus* *sciuri*	*Escherichia coli*	*Pseudomonas* *aeruginosa*	*Aspergillus flavus*	*Penicillium* *chrysogenum*
1	ND	ND	ND	ND	ND	ND
2	ND	17	39	24	ND	ND
Ceftriaxone	13	12	11	11		
Natamycin					18	19

**Table 6 ijerph-17-01377-t006:** Antifungal activity of the treated polyester dyed fabrics with TiO_2_ NPs.

Dye No.	*Aspergillus Flavus*	*Penicillium Chrysogenum*
1	21	19
2	ND	ND
Natamycin	18	19

(ND) Not Detected.

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
