# Peer review of "Nano TiO2 Imparting Multifunctional Performance on Dyed Polyester Fabrics with some Disperse Dyes Using High Temperature Dyeing as an Environmentally Benign Method"

_ijerph, 2020, doi:10.3390/ijerph17041377_

Round 1
Reviewer 1 Report
Comments to authors
This MS reported multifuntional TiO2-nanoparticles coated dyed polyester fabrics by a environmental high temperature dyeing menthod, the as-prepared funtional polyester fabric with various concentrations of TiO2 nanoparticles exhibited good self-cleaning, light-fastness, UV shielding and anti-fungi properties. It is an interesting work but it might be accepted after the clear statement of high novelty of this work to others. In addition, some poor discussion and errors can be found during the whole writing. The details can be found as follows:
In the Materials and Methods part, the used chemicals including purity and manufacturer in this paper should be added. The author introduced the dyeing process, but didn’t mention how to prepare TiO2-NPs treated polyester fabric. In 2.2 color measurement, the meaning of R, R0 in the K/S value equation should be clarified. In 2.3 Fastness property test, please describe simply the test methods and evaluation criteria of each fastness property test standard. How about the UV blocking property for pristine TiO2 or PDMS coated blank polyster fabric? Why choose TiO2 because there are many kinds of semiconductors, e.g. ZnO or CuO, could realize similar goal. In 3.2 fastness property, what does the “Alteration” mean? How many levels for both of the fastness properties and what level of number 5 is? In 3.4 self-cleaning of post-treated part, what’s the concentration of Methylene Blue? Do you have optical images of untreated and treated polyester fabrics before and after UV illumination. Some recent highly related references about multifunctional TiO2 particles coated fabrics should be mentioned and updated, such as J. Mater. Chem. A, 2017, 5(1), 31-55; Materials & Design 2017, 128, 1-8. Please supplement SEM images of polyester fabrics before and after treatment with TiO2-NPs and characterize the crystal type of TiO2-NPs treated polyeaster fabric. In the 3.6 antimicrobial activities part, the authors qualitative characterized the antimicrobial activities of untreated dyed polyester fabric against bacterials and fungi, sample No.2 showed strong antibacterial property against Staphylococcus sciuri, Escherichia coli and Pseudomonas aeruginosa, what’s the antibacterial mechanism? Did the authors characterize the antibacterial activities of treated dyed polyester fabric? As what the authors analysed, inorganic oxides NPs (TiO2) may endow the fabric with antimicrobial property, the sample No.1 exhibited good antifungal activity, but sample No.2 does not show any antifungal activity, why? There are many grammatical and spelling mistakes in this MS and some confused sentences which make it hard to follow. Such as “The hydrophobic surface absorbs dirt, keeping the polyester outer surface constantly clean”.
Reviewer 2 Report
The authors have shown that post treatment of dyed polyester fabric with a nano-particle titania gives rise to some useful anti-fungal, anti-microbial activities. Although the science is interesting and useful outcomes I do have some concerns with regard to the authors methods. The activities described by the authors is not new and quite well known in the literature. My concerns here are :
What is the type of nano-titania here being used-anatase and what are it properties and how was it made. This is an unknown entity in the article and without such information the paper is useless. How was the nano-titania applied to the fabric? What was the post wash permanence of the fabric on activity? If this activity is lost after one post-wash then the article is of no interest at all. All these points must be addressed before publication. The grammar in parts needs attention.Author Response
Please see the attachment

Reviewer 3 Report
The quality of the manuscript is not satisfactory at this current state. The authors please see the following comments:
1- The abstract of the paper is not complete enough. It should highlight the main achievements of the study.
2- The English level should be improved. Some sentences are very difficult to understand.
3- The methodology part should be completed. The dyeing process should be explained more clearly. The treatment method of samples with TiO2 should be explained in detail.
4- The method of conducting the antimicrobial test should be explained in detail and the related standards should be provided.
5- The novelty of the work should be explained in Abstract and Introduction.
6- What is the role of TiO2 nanoparticles?
7- How stable are TiO2 nanoparticles on polyester?
Round 2
Reviewer 1 Report
The authors have implemented the suggested changes and responded in detail. The paper is now suitable for publication.
Author Response
Thank you for your time, efforts and help
Reviewer 2 Report
Thanks you for the detailed revisions thats fine now
Author Response
Thank you for your time, efforts and help.
Reviewer 3 Report
The abstract and introduction of manuscript is not clear and informative. The novelty of this work is not still clear and it needs a major revision in terms of English writing style. No sufficient images and characterisations have been provided.
